# Numerical Analysis of Electromagnetic Field Exposure from 5G Mobile Communications at 28 GHZ in Adults and Children Users for Real-World Exposure Scenarios

**DOI:** 10.3390/ijerph18031073

**Published:** 2021-01-26

**Authors:** Maria Sole Morelli, Silvia Gallucci, Beatrice Siervo, Valentina Hartwig

**Affiliations:** 1U.O.C. Infotel, Fondazione Toscana Gabriele Monasterio, 56100 Pisa, Italy; msmorelli@monasterio.it; 2DIITET Department, Institute of Electronics and Information and Telecommunications Engineering IEIIT-CNR, 20133 Milan, Italy; silvia.gallucci@polimi.it; 3Department of Electronics, Information and Bioengineering (DEIB), Politecnico di Milano, 20133 Milan, Italy; 4Department of Information Engineering, University of Pisa, 56100 Pisa, Italy; siervo.beatrice@libero.it; 5DSB Department, Institute of Clinical Physiology IFC-CNR, 56100 Pisa, Italy

**Keywords:** 5G, mmW, numerical dosimetry, smartphone exposure

## Abstract

The recent development of millimeter-wave (mmW) technologies, such as the fifth-generation (5G) network, comes with concerns related to user exposure. A quite large number of dosimetry studies above 6 GHz have been conducted, with the main purpose being to establish the correlation between different dosimetric parameters and the skin surface temperature elevation. However, the dosimetric studies from 28 GHz user equipment using different voxel models have not been comprehensively discussed yet. In this study, we used the finite-difference time-domain (FDTD) method for the estimation of the absorption of radiofrequency (RF) energy from a microstrip patch antenna array (28 GHz) in different human models. Specifically, we analyzed different exposure conditions simulating three real common scenarios (a phone call scenario, message writing scenario, and browsing scenario) regarding the use of smartphones/tablets by four different individuals (adult male and female, child male and female). From the results of Absorbed Power Density (Sab), it is possible to conclude that all the considered exposure scenarios comply with the safety limits, both for adult and children models. However, the high values of the local Specific Absorption Rate (SAR) in the superficial tissues and the slight differences in its distribution between adults and children suggest the need for further and more detailed analysis.

## 1. Introduction

In recent years, increasing demands for greater channel capacity and higher data rates for mobile communication has given rise to exploration of the millimeter-wave (mmW) spectrum towards the fifth-generation (5G) network system.

5G technologies will allow higher data transfer rates and user densities at less power per bit, permitting the development of future wireless communication technology for handsets, tablets, and computers to the Internet of Things (IoT) applications [1,2]. 5G includes communication systems with a higher operating frequency than those used for previous technologies such as the fourth generation (4G) network or the Long Term Evolution (LTE) network. The bands assigned to this emerging system are very large and include frequencies that vary from below 6 GHz to dozens of GHz: among these frequencies, the band above 28 GHz is currently used in the USA for the upcoming 5G systems [3,4]. Also, the European Commission has recently adopted an Implementing Decision to harmonize the radio spectrum in the 24.25–27.5 GHz (or 26 GHz) band [5].

Although the radiation involved in 5G technology is non-ionizing (radio frequency (RF) waves), with the growing development of systems based on mmW technologies, it has become important to evaluate any harmful effects on health caused by electromagnetic fields [6]. There is a dire need to study the absorption of electromagnetic energy in human tissues due to exposure to mmW radiation.

To protect human bodies from exposure to mmW radiation, many countries set safety guidelines standards, such as those set by the International Commission on Non-Ionizing Radiation Protection (ICNIRP) [7] and the Institute of Electrical and Electronics Engineers (IEEE) [8], as well as the Federal Communications Commission (FCC) [4].

Most of the studies about the exposure assessment of the human body to RF radiation in complicated exposure conditions used numerical approaches such as numerical analysis based on the finite-difference time-domain (FDTD) method [9,10,11,12,13]. A quite large number of dosimetry studies above 6 GHz have been conducted [14,15,16,17,18,19,20,21,22], with the main purpose to establish the correlation between different dosimetric parameters such as the specific absorption rate (SAR), absorbed power density, and the skin surface temperature elevation.

In 2017 Guraliuc et al. [18] presented a detailed numerical dosimetry study with a 60 GHz antenna module for two representative human body exposure scenarios within 5G small cells. They used the numerical model of an adult human head together with a human hand that held a smart-phone. For the numerical analysis, the finite integration technique was implemented together with a finite element method used to better characterize the antenna. As dosimetric parameters, SAR absorbed power and equivalent incident power density was used. The finding of this study was that the maximum absorption occurs on the surface of the user’s ear (for the phone call scenario) and fingertips (for the browsing scenario).

In 2018, Hamed and colleagues [19] investigated the SAR distributions due to a radiating source antenna using the FDTD method in single and layered human tissues by evaluating the SAR 1g and point SAR (without mass averaging) at frequencies of 28, 40, and 60 GHz. The results concluded that at the radiated power of 20 and 24 dBm, SAR levels (without mass averaging) in the tissues at 28 GHz were less than 40 and 60 GHz.

Recently, Vilagosh and colleagues [21] investigated pulsed 30–90 GHz radiation penetration into the human ear canal and tympanic membrane and concluded that there is very low penetration of RF radiation into the ear canal at 30 GHz and that, in this specific exposure condition, it is not likely to have a significant thermal impact on the tympanic membrane.

Gultekin et al. [22] examined RF exposure and absorption in ex vivo bovine brain tissue and a brain simulating gel at three frequencies (1.9 GHz, 4 GHz, and 39 GHz) that are relevant to 4G and 5G spectra. They found that the radiation penetration in the brain tissue, and therefore SAR, decreases with increasing depth and frequency.

Despite the extensive literature on the subject [8,10,23,24,25,26], there has been long-standing controversy about whether children absorb more RF energy than adults when using a smartphone [27]. Using realistic virtual human models, it is possible to find any differences in the exposure for different users such as adults vs. children. Moreover, different representative and usual scenarios should be considered, beyond the classic phone call scenario, to have a wider knowledge of the interaction between mmW and human tissues.

The main purpose of this study is a numerical analysis of the local SAR distribution within different realistic 3-D human models exposed to the electromagnetic field of a mobile-communication antenna at the frequency of the latest generation of mobile networks 5G.

We calculated, using the FDTD method, the local SAR in different exposure conditions by simulating three real common scenarios (phone call scenario, message writing scenario, and browsing scenario) regarding the use of smartphones/tablets by four different individuals (adult man and woman, child male and female).

## 2. Materials and Methods

### 2.1. Antenna Model

Microstrip antennas are mostly considered suitable for modern smartphones due to their low cost, low profile, simple fabrication, and easy replication for large arrays [28]. The 2 × 2-patch single-layer antenna array (at 28 GHz) described in [19] was used.

Briefly, the antenna patch was made of copper (thickness = 0.0035 mm), the ground layer (with infinitesimal thickness) was chosen as a Perfect Electric Conductor (PEC), and as a dielectric, a Roger RT/Duroid 5880 material (thickness = 0.268 mm, εr = 2.2, loss tangent = 0.0009, σ = 00308 S/m) was chosen. A sinusoidal voltage source (@28 GHz, 50 Ω) was placed on the patch surface as feed. The coefficient reflection S11 was −14.29 dB, and VWSR = 1.48. The efficiency of the antenna was about 89.45% and its gain was 13.83 dBi. Present ICNIRP guidelines do not mention total radiated power for microstrip antennas, so the input power level for the used antenna was chosen from the literature review [20,29,30,31]. Moreover, the 5G mobile communication specifications by the 3rd Generation Partnership Project (3GPP) indicate a maximum total radiated power (TRP) for commercial user equipment (UE) products (power class 3: handheld) of 23 dBm at 28 GHz [32].

Hence, the total input power level was set to 15 dBm (@32 mW), which is a reasonable estimate for 28 GHz.

The microstrip antenna was placed in a typical smartphone case to evaluate the real radiation pattern: the dielectric material constituting the case was Polyvinyl chloride (εr = 2.96, loss tangent = 0.01, σ = 0.016 S/m). The antenna was placed 2 mm inside the smartphone case, in the upper right corner of the smartphone front [18]. In Figure 1, the patch antenna is shown in the smartphone and all dimensions are listed.

### 2.2. Exposure Scenarios

To simulate some real and common exposure conditions during the daily use of a smartphone/tablet, we chose three different exposure scenarios with different positions of the radiation terminal with respect to the human model, as shown in Figure 2.

The expected usages are the following:
Phone call scenario: the smartphone is close to the right ear of the model. The phone case was in contact with the ear so the antenna-skin distance was 2 mm.Message writing scenario: the terminal in front of the chest. The phone case was in line with the beginning of the sternum (jugular notch).Browsing scenario: the terminal in front of the abdomen. The phone case was in line with the last ribs. This scenario simulates the use of a smartphone to consult social applications.


Distances between human models and antenna are shown in Table 1, with reference to Figure 2. The terminal was placed with the typical angle aligned to the vertical (@20°).

### 2.3. Numerical Simulations

Numerical analysis was performed using the FDTD method [33] implemented in XFdtd (Remcom, State College, PA, USA). An automatic non-uniform mesh was chosen with a minimum size of 0.08 mm and a maximum size of 1.071 mm. The maximum cell size was automatically chosen to be equal to one-tenth of the free space wavelength in compliance with the well-known rule to suppress the numerical dispersion error in FDTD simulations [34,35]. Using the intelligent meshing option, we set the minimum number of grid cells per wavelength to be equal to 15. The material properties of dielectric materials, such as human tissues, have been used to determine the wavelength. As a result of this automatic meshing, the human model’s discretization had a resolution of 0.15 mm in the region of interest. The adopted resolution enabled obtaining good computational accuracy while also considering the penetration depth of radiation in the skin, which is about 1 mm at 28 GHz. The convergence of the results was reached with a threshold of 30 dB. Due to the computational limit, to reduce the total mesh size, the non-exposed part of the human models was not included in the mesh. Specifically, for all the human models, in the phone call scenario, the mesh extended from the crown of the head to the jugular notch, in the message writing scenario from the crown of the head to the line of last ribs, and in the browsing scenario from the jugular notch to the genitals (see Figure 2).

We used perfect matched layers (seven layers) [33] with 20 cells of free space surrounding the voxel domain as boundary conditions.

All of the simulations were performed using a workstation Intel Core i5 @2.67 GHz, RAM 4 GB (Intel Corporation Italia spa, Milan, Italy), equipped with a graphical card NVidia Geforce GTX 750 GPU (NVIDIA Corporation, Santa Clara, CA, USA) and Windows 10 (Microsoft Italia, Milan, Italy) operating system.

Four available realistic human models belonging to the Virtual Population [34] were used in our simulations: Duke (adult male, age 34, height 1.77 m, mass 70.2 kg, BMI 22.4 kg/m^2^, 146 tissues), Ella (adult female, age 26, height 1.63 m, mass, 57.3 kg, BMI 21.5 kg/m^2^, 76 tissues), Billie (pre-teenager female, age 11, height 1.49 m, mass 34.0 kg, BMI 15.4 kg/m^2^, 112 tissues), and Thelonious (male child, age 6, height 1.15 m, mass 18.6 kg, BMI 14.1 kg/m^2^, 76 tissues).

All these models were based on high-resolution magnetic resonance (MR) images of healthy volunteers.

Electric properties (dielectric constant and conductivity) of the tissues included in each model were calculated at 28 GHz according to the Cole-Cole method [35]. All the models were in a standing position with arms by their side.

For each human model, all the three exposure scenarios described previously were simulated, with a total number of simulations performed of 12.

### 2.4. Data Processing

For each situation of exposure, local SAR in the entire model was estimated, that is the SAR value for each grid cell, calculated as the ratio between the absorbed power in each grid divided by grid mass. Then, we found the maximum local SAR of all the grid cells for each human model.

A radiofrequency electromagnetic wave exponentially decays from the surface to deeper regions in the human body [6]. According to this, SAR at a certain depth in the tissues can be calculated from the knowledge of the SAR value at the surface of the skin and the so-called penetration depth or skin depth. This quantity is conventionally defined as the depth corresponding to the SAR of 1/e^2^ (or 13.5%) of SAR at the surface [6,27].

For the phone call scenario, the SAR penetration depth along the zdirection, from the surface (ear skin) to the deeper regions, has been calculated for all the models.

The SAR trend along the zaxis has been evaluated from the axial SAR map extracted from FDTD results, while considering the cell with the maximum value of the local SAR.

Moreover, the SAR at a 1 mm depth (by percentage with respect to its peak value) in the head model was also calculated.

The most recent ICNIRP guidelines [7] give Basic Restrictions for frequencies >6 GHz to 300 GHz in terms of Absorbed Power Density (Sab) rather than SAR. To obtain a direct understanding of the safety of the chosen exposure scenario, we calculated Sab according to the ICNIRP definition:
(1)$$Sab=\iint_{A}^{}dxdy\int_{0}^{zmax}\rho \left(x,y,z\right)\cdot SAR\left(x,y,z\right)dz/A,$$
where the body surface is provided at *z* = 0, *zmax* is the depth of the body at the corresponding region, and *A* = 0.02 m × 0.02 m represents the averaging area. The given limit for *Sab* is 20 W/m^2^.

## 3. Results

Table 2 shows the numerical values of the computed parameters. The local SAR is shown as 2D maps for each plane (axial, coronal, and sagittal plane) using a color scale (Figure 3). Values are in dB relative to the maximum value estimated in the model. The color range is between 0 dB (red) and −350 dB (dark purple).

To better visualize the SAR distribution, in the representation of the phone call scenario, the smartphone is not included.

### 3.1. Phone Call Scenario

Since the very low distance between the antenna and the head (2 mm), the maximum value of local SAR was significant in all the human models (Table 2), but only in the superficial tissue (limited to the ear skin), as shown in Figure 3. The maximum local SAR value was in alignment with the ear apex for Duke, Ella, and Billie models and was also in alignment of the tragus for Thelonious.

All of the obtained Sab values were well below the safety limit imposed by the most recent ICNIRP guidelines, that is 20 W/m^2^.

Billie had the highest value for the local SAR and Sab. Moreover, Ella had the lowest value of the local SAR, which was comparable to the maximum local SAR of Duke. The SAR distribution in the head shows that only a very small portion of the ear skin presented a high value of SAR for all the models. From the axial view, the power absorption was limited to the superficial tissue in the head for all the human models. The local SAR fell out very quickly for all models. For example, the SAR value fell out of 42 dB with respect to the maximum value in the ear cartilage for the Duke model. As reported in Table 3, the SAR depth of penetration was lower than 1 mm for Ella, Billie, and Thelonious models and was equal to 1.77 mm for Duke. Interestingly, Duke had the highest value of SAR depth of penetration to indicate a deeper power absorption in the male adult head with respect to the child’s head, although this was limited to the superficial tissue. In effect, at 1 mm depth, the SAR value was about 2% with respect to its peak value for Ella, Billie, and Thelonious models, and 32.33% for the Duke model. Figure 4 shows the local SAR values (dB) along with the profile of maximum SAR value for the whole head, from the right ear to the left ear. In Figure 5, the local SAR value (normalized to the maximum value at the surface of the skin) is shown in a zoom area relative to the first 2 mm depth from the skin surface, for each model.

### 3.2. Message Writing Scenario

As expected, maximum local SAR values were lower with respect to the previous scenario, since the greater distance was present between the antenna and the human models (157 mm). For this scenario, Ella had the highest value of local SAR and Sab.

Looking at the SAR maps (Figure 6) it is possible to note that the higher values were located in the facial area, despite the greater distance between the antenna and the face with respect to the previous scenario, and also in the chest area for all the models. For Duke, Ella, and Thelonious models, the maximum local SAR value was in alignment with the lower part of the nose. In contrast, Billie had the maximum local SAR value on the chest area. This could be due to the slightly different posture of the model.

Even in this scenario, the highest local SAR values were confined in the skin: inner tissues (mucosa, cartilage, connective tissue, subcutaneous adipose tissue, muscle, etc.) had a SAR reduced of more than 10 dB with respect to the peak value on the skin.

### 3.3. Browsing Scenario

In this last scenario, the maximum dosimetric parameter values were the smallest ones due to the greater distance between the antenna and the human models (207 mm).

Again, Ella had greater values of local SAR and Sab, with respect to the other models.

In this exposure configuration, the face was no longer affected by the power absorption for all the models, while the entire thorax area had high local SAR values (Figure 7). The maximum value of local SAR, which was always limited to the skin, was in alignment with the abdomen for Duke, Ella, and Thelonious and to the lower abdomen for Billie. For the Duke model, high local SAR values were also found in the skin of the genital area, although the SAR value of the penis had fallen by 60 dB with respect to the peak value.

## 4. Discussion

The 5G technology for next-generation mobile communications is arousing a lot of interest since it permits greater channel capacity and higher data rates. The bands assigned to this emerging system includes the 28 GHz band, which is currently used in the USA for the 5G systems [3,4], and the 26 GHz band, which was recently adopted by the European Commission [5].

This paper presents a numerical analysis of the human body exposure to a 5G generic mobile terminal operating at mmW, considering different users and different exposure scenarios. We presented a numerical analysis of the human body’s exposure to a 5G 2 × 2-patch single-layer antenna array operating at mmW (@28 GHz, 30 mW), to be used in a commercial smartphone or tablet. Three representative realistic scenarios (phone call, message writing scenario, and browsing scenario) were considered and four different human virtual models were used in the numerical simulations. We numerically computed the local SAR and calculated the relative Sab, since it is the dosimetric quantity adopted by the relevant guidelines.

Since very low values were obtained for Sab (<2.15% of ICNIRP basic restriction in the phone call scenario, <0.16% of ICNIRP basic restriction in the writing scenario, and <0.03% of ICNIRP basic restriction in the browsing scenario), we can conclude that all the considered exposure scenarios comply with the safety limits relative to RF exposure, both for adult and child models [4,7,8]. However, in specific and highly localized superficial points of the human models, the local SAR peak value was significant. This becomes important for some human tissues which are particularly vulnerable to electromagnetic radiation-induced heating: for example, the eyes that, in addition to be located on the body surface, are less perfused to blood so they cannot properly redistribute the generated heat [6].

From our calculations of the SAR depth of penetration, it was demonstrated that the RF power absorption decays very quickly in depth. Since the thickness of the human epidermis and dermis is about 0.1 and 2 mm, respectively, we can conclude that the mmW energy is rapidly absorbed within the epidermis and dermis and does not reach the deeper tissues in the body [36].

This shows again, in agreement with previous literature [18,19,21,22,37], that at this frequency and with the incident power values typical of the smartphone technology, it is highly unlikely that inner tissues absorb health-harmful power levels.

From a comparison with a very similar exposure scenario including a 4G smartphone [10], we found that at 28 GHz, which is one of the frequencies intended for use in 5G technology, the reduction of SAR from the surface to the innermost tissues is much more sudden. As demonstrated in this study, in fact, the SAR value at a depth of 1 mm in the head is about 30% of the surface value in the adult male model and is about 2% of the surface value in adult female and children models. Moreover, in the present study, we did not find any particular hot spots, as were reported in the 4G study taken in comparison.

Regarding the question about whether children absorb more RF energy than adults when using a smartphone, the debate is still heated in the literature and to date, there are no firm conclusions on this matter [23,24,25,26,27]. The differences in peak local SAR and Sab between adults and children also found in this work are not relevant for compliance assessment [18] but might be important for other purposes, including research on possible biological effects of RF energy. Anyway, assuming that there are any differences in power absorption distribution between adults and children, they could be relevant at lower frequencies (@4G and previous communication technologies) but would have minimal relevance at mmW frequencies since, as shown in this study, the absorption is exclusively localized on the skin for all the users. Interestingly, we found that Duke had the highest value of SAR depth of penetration to indicate a deeper power absorption in the adult male head with respect to the child’s head, limitedly to the superficial tissue. In light of this, further studies can be suggested regarding the localized penetration of mmW radiation into specific tissue of human with different age: for example, since is well known that as the skin ages, it becomes thinner and more fragile [37], it could be interesting to study the power absorption distribution into the human ear canal and tympanic membrane of elderly 5G smartphone users [21]. Moreover, since a higher water content in the tissue leads to high energy absorption [6], elderly skin with lower water content is potentially more easily penetrated by the mmW energy [36].

### Limitations and Further Studies

A limitation of this study regards the posture of the human models which were in a standing position with arms by their side. This fact did not reflect reality as for all the three simulated scenarios, the smartphone should be handheld with the arms being bent.

The absolute value of the local SAR is strongly dependent on the relative position of the antenna with respect to the human model, so small (millimeter) variations in the position of the antenna will result in significant changes in the SAR.

We kept the same antenna-head relative position for all models but, due to the different head shape of the models, the angles of incidences of RF radiation on the skin were probably different. Further studies should be performed to clarify this point.

On the other hand, since many variables determine the SAR during realistic usages, a generalization to human populations under real-world exposure conditions is very complex.

## 5. Conclusions

This paper presents a numerical analysis of the human body’s exposure to a 5G generic mobile terminal operating at mmW, considering different users and different exposure scenarios. For all the considered exposure scenarios, the results show very low values for the Sab. From calculations of the SAR depth of penetration, it is demonstrated that the mmW energy is rapidly absorbed within the epidermis and dermis. Some slight differences in peak local SAR between adults and children were found in this work, but they have minimal relevance at mmW frequencies, since the absorption is exclusively localized on the skin for all of the users.

## Figures and Tables

**Figure 1 ijerph-18-01073-f001:**
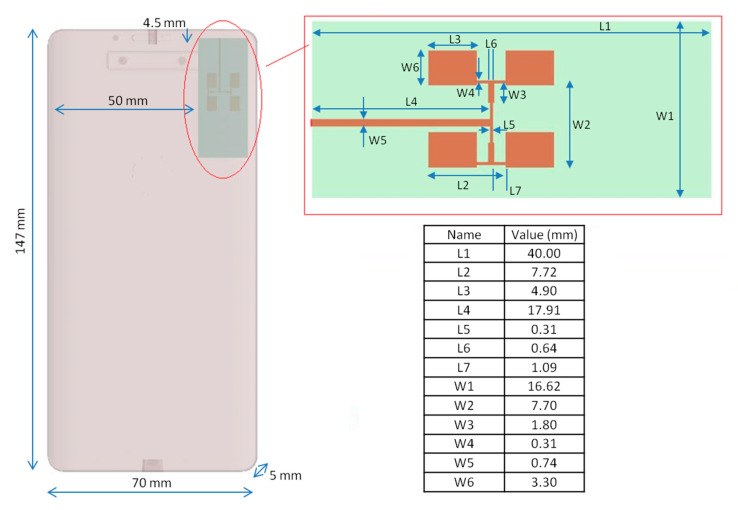
Smartphone with the antenna integrated and the size of the antenna elements.

**Figure 2 ijerph-18-01073-f002:**
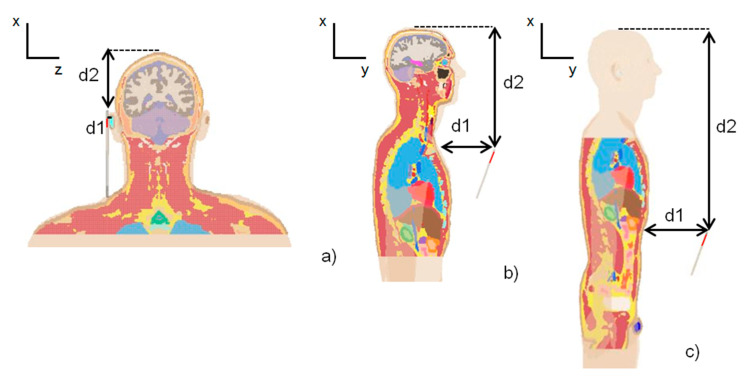
Exposure scenarios: (**a**) phone call, (**b**) message writing, (**c**) browsing.

**Figure 3 ijerph-18-01073-f003:**
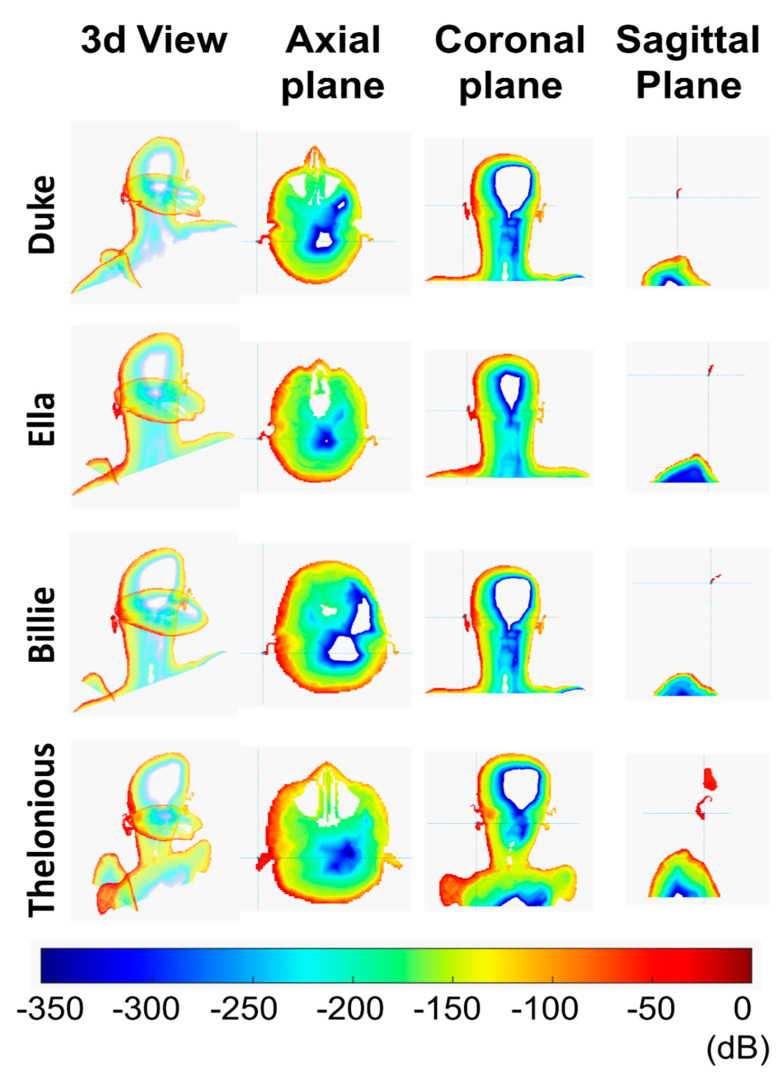
Specific Absorption Rate (SAR) distribution in a phone call scenario in three different planes: axial, coronal, and sagittal.

**Figure 4 ijerph-18-01073-f004:**
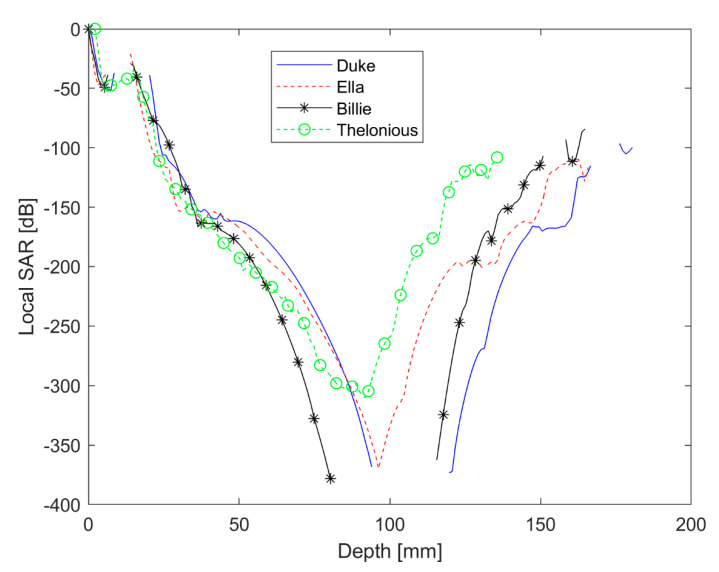
Local SAR values (dB) along the profile of the maximum SAR value.

**Figure 5 ijerph-18-01073-f005:**
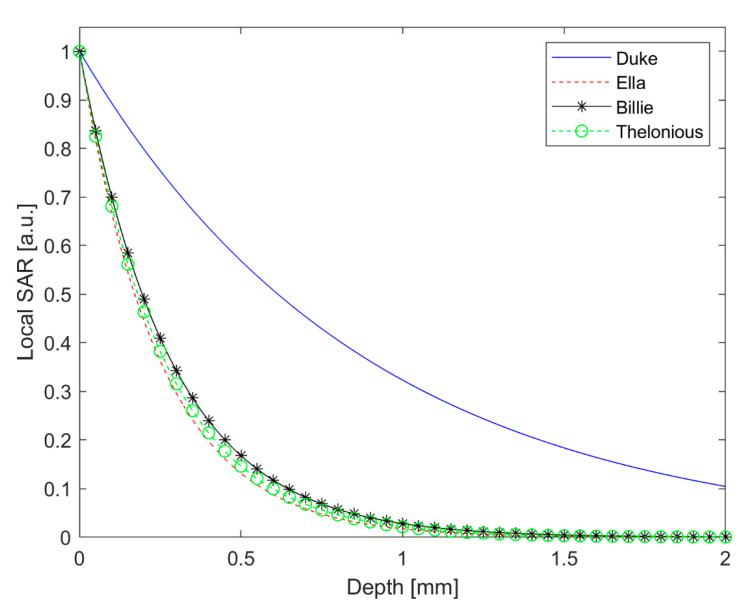
Local SAR (a.u.) for the first 2 mm depth in the head.

**Figure 6 ijerph-18-01073-f006:**
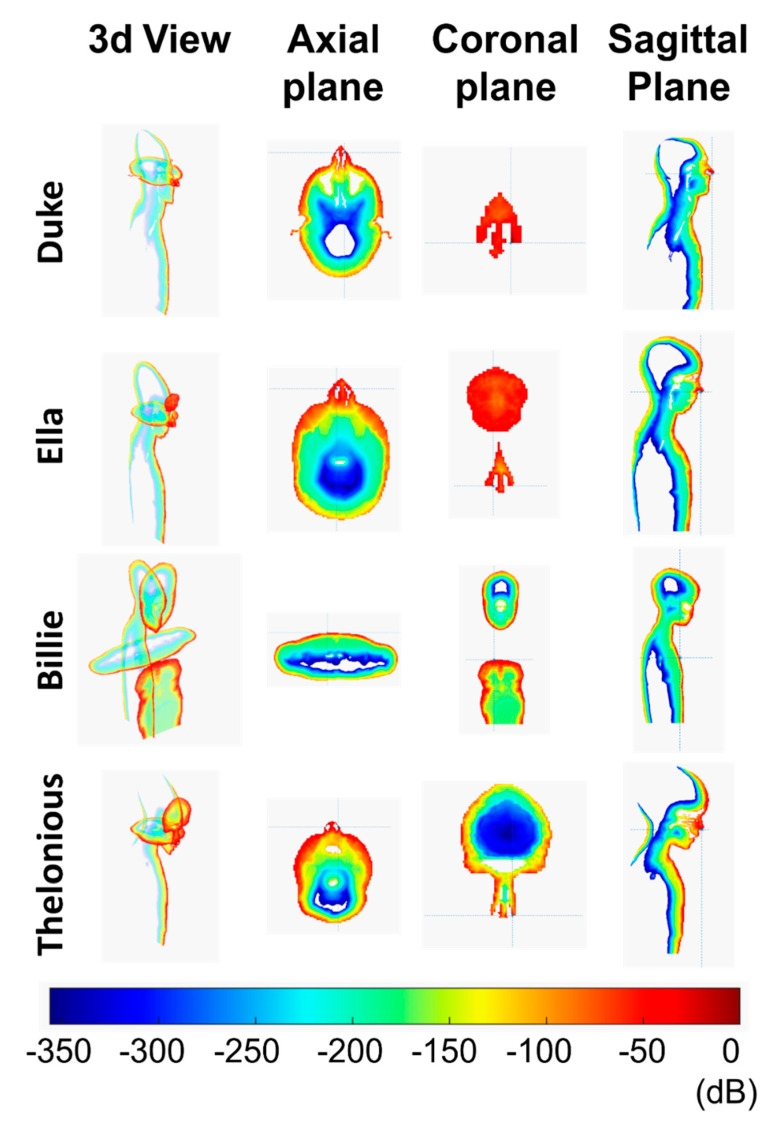
SAR distribution in the message writing scenario in three different planes: axial, coronal, sagittal.

**Figure 7 ijerph-18-01073-f007:**
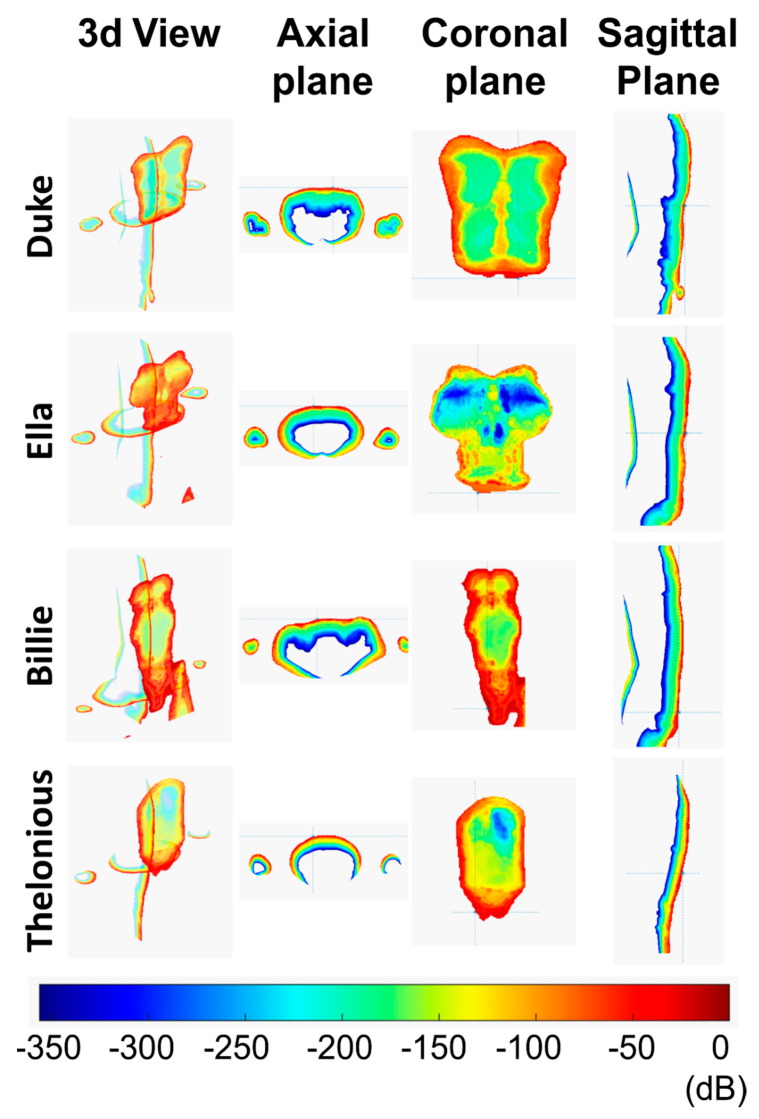
SAR distribution in the browsing scenario in three different planes: axial, coronal, and sagittal.

**Table 1 ijerph-18-01073-t001:** Human models-antenna distances for each exposure scenario.

	Phone Call		Message Writing		Browsing	
	d1 (mm)	d2 (mm)	d1 (mm)	d2 (mm)	d1 (mm)	d2 (mm)
Duke	2	124	157	346	207	642
Ella	2	121	157	317	207	596
Billie	2	110	157	281	207	541
Thelonious	2	108	157	241	207	457

**Table 2 ijerph-18-01073-t002:** Numerical results of finite-difference time-domain (FDTD) simulations.

	Phone Call		Message Writing		Browsing	
	Max Local SAR (W/kg)	Sab (W/m^2^)	Max Local SAR (W/kg)	Sab (W/m^2^)	Max Local SAR (W/kg)	Sab (W/m^2^)
Duke	2.82	0.13	0.17	8.30 × 10^−3^	0.02	2 × 10^−3^
Ella	2.60	0.05	0.37	32.10 × 10^−3^	0.06	6.10 × 10^−3^
Billie	12.75	0.43	0.12	11 × 10^−3^	0.02	2.20 × 10^−3^
Thelonious	8.44	0.21	0.10	14.60 × 10^−3^	7.15 × 10^−3^	0.5 × 10^−3^

**Table 3 ijerph-18-01073-t003:** SAR depth of penetration and SAR at 1 mm depth (in % with respect to the peak value) for the phone call scenario.

	SAR Depth of Penetration (mm)	SAR at 1 mm (%)
Duke	1.77	32.33
Ella	0.49	1.73
Billie	0.56	2.81
Thelonious	0.52	2.14

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
