# Peer review of "Numerical Analysis of Electromagnetic Field Exposure from 5G Mobile Communications at 28 GHZ in Adults and Children Users for Real-World Exposure Scenarios"

_ijerph, 2021, doi:10.3390/ijerph18031073_

Round 1

Reviewer 1 Report

This is a useful modelling paper to estimate the induced fields in the skin of users of 5G devices. Numerical dosimetry is notoriously difficult in near-field conditions, which this is (wavelength in air at 30 GHz is approx. 1 cm). The modelling uses the well-known package xFDTD and selected child and adult phantoms from the ‘Virtual Family’. In the main, it is well written, but there does need to be some careful proofreading for English usage. There are some major issues which need addressing:

  1. The paper references ICNIRP 2020 (7), which gives Basic Restrictions for the frequency chosen (28 GHz) in terms of Absorbed Power Density (Sab) rather than SAR for local exposures. xFDTD permits the estimation of this quantity following the recommendations on p504 of this reference. It would be useful to express the results of this simulation as a percentage of the Basic Restrictions. The distribution of SAR within the skin, although useful, does not really give an insight into possible temperature elevation, due to heat convection by capillary blood and other diffusive processes. This should be added to the discussion and possibly some additional simulations carried out.
  2. The assumption of 32 mW on p 3(or 32 mW on p 5) for the total radiated power level needs further discussion. I am unable to find in reference (7) any mention of 18 dBm as an ICNIRP recommendation. Further, ref 29 is referring to the ‘old’ 1998 ICNIRP standard rather than the updated one (and this standard doesn’t mention total radiated power either). There are some assumptions in ref 29 in deriving the 18 dBm which need further discussion and the actual figure needs to reflect reference (7) rather than the ‘old’ standard.
  3. The results shown in Figure 6 seem remarkable: that adult females should have a local SAR value around 25% of adult males (and less than female children). There is no mention of this in the discussion that I can see (I may have missed it). Whether or not it has to do with different fat composition in the skin of different models needs further discussion. It may be worth repeating the simulation to eliminate possible artefact.
  4. A minor point, but the definition of penetration depth varies, depending on whether power or field value is referred to. Here it’s field value and is consistent with ref (7) p 508 (86% absorbed or 14% remaining, with the field value 37% of original). This needs to be made clear. Incidentally, this may not hold for near-field conditions (see eq 6 on p 504 of ref (7)).

Grammar needs attention: for example:

P 1. The bands assigned to … includes frequencies below..’

P 2. ‘human hand to represent the smart-phone model.’\

‘simple tissue models estimation of the differences…’

P 3. Antenna patch was constituted of copper..

‘The antenna was inserted of about 2 mm..’ – meaning unclear

P 11. ‘the thorax was entirely interested…’, the genital area was also interested…’ – not clear what is meant by ‘interested’

P 15. ‘mobile terminal’ maybe ‘mobile device’

Author Response

The authors want to thank the reviewers for their precious comments about the manuscript.

The manuscript is substantially changed and it was rewritten. For this reason, the reviewers can find both a marked version and a clean version of the manuscript. Please, refer to the marked version for the point-by-point response to reviewers.

Reviewer 1

1. The paper references ICNIRP 2020 (7), which gives Basic Restrictions for the frequency chosen (28 GHz) in terms of Absorbed Power Density (Sab) rather than SAR for local exposures. xFDTD permits the estimation of this quantity following the recommendations on p504 of this reference. It would be useful to express the results of this simulation as a percentage of the Basic Restrictions. The distribution of SAR within the skin, although useful, does not really give an insight into possible temperature elevation, due to heat convection by capillary blood and other diffusive processes. This should be added to the discussion and possibly some additional simulations carried out.

As suggested by both the reviewers, we added in the manuscript the results of our simulations in terms of Absorbed Power Density (see new Table 2). We explained, in a dedicated section “Data processing” in the Materials and Methods section, how Sab has been calculated and reported the results in the new Table 2. Moreover, in Table 2 we eliminated the data relative to averaged SAR and total power dissipated (see new Table 2), since they are not relevant for our Discussion.

We changed, in the Discussion section (page 17 of 20), our comment relative to the compliance of the simulated exposure conditions with the safety limits:

Since the very low values obtained for Sab (<2.15% of ICNIRP basic restriction in the phone call scenario, <0.16% of ICNIRP basic restriction in the writing scenario, and <0.03% of ICNIRP basic restriction in the browsing scenario) , we can conclude that all the considered exposure scenarios comply with the safety limits relative to RF exposure, both for adult and child models [4,7,8].

2. The assumption of 32 mW on p 3(or 32 mW on p 5) for the total radiated power level needs further discussion. I am unable to find in reference (7) any mention of 18 dBm as an ICNIRP recommendation. Further, ref 29 is referring to the ‘old’ 1998 ICNIRP standard rather than the updated one (and this standard doesn’t mention total radiated power either). There are some assumptions in ref 29 in deriving the 18 dBm which need further discussion and the actual figure needs to reflect reference (7) rather than the ‘old’ standard.

As the reviewer states, present ICNIRP guidelines do not mention total radiated power for microstrip antennas. The input power level for the used antenna was chosen from literature review, and also from the 5G mobile communication specifications by the 3rd Generation Partnership Project (3GPP) that indicates a maximum total radiated power (TRP) for commercial user equipment (UE) products (power class 3: handheld) of 23 dBm at 28GHz [32]. Hence, the total input power level was set to 15 dBm (@32mW), which is a reasonable estimate for 28 GHz. We changed the sentence as following (page 3 of 20):

Present ICNIRP guidelines do not mention total radiated power for microstrip antennas, so the input power level for the used antenna was chosen from literature review [20,29–31]. Moreover, the 5G mobile communication specifications by the 3rd Generation Partnership Project (3GPP) indicate a maximum total radiated power (TRP) for commercial user equipment (UE) products (power class 3: handheld) of 23 dBm at 28GHz [32]. Hence, the total input power level was set to 15 dBm (@32mW), which is a reasonable estimate for 28 GHz.

3. The results shown in Figure 6 seem remarkable: that adult females should have a local SAR value around 25% of adult males (and less than female children). There is no mention of this in the discussion that I can see (I may have missed it). Whether or not it has to do with different fat composition in the skin of different models needs further discussion. It may be worth repeating the simulation to eliminate possible artefact.

We thank the reviewer for this comment. The simulation for adult female has been repeated but the results did not change, so we think there is not any artefacts. However, we checked all the data reported in the manuscript and we found an error in adult male results for phone call scenario. So, we corrected the SAR value for Duke model in phone call scenario (see new Table 2) and, consequently, the relative data for SAR penetration depth and SAR at 1 mm (see new Table 3).

From the corrected data, local SAR values for adult female and adult male are now comparable and lower than ones of children.

We added some considerations about this point in the Discussion (page 18 of 20):

The absolute value of local SAR is strongly dependent from the relative position of the antenna respect to the human model, so small (millimeter) variations in position of the antenna will result in significant changes in SAR.

We kept the same antenna-head relative position for all models but, due to the different head shape of the models, the angles of incidences of RF radiation on the skin were probably different. Further studies should be performed to clarify this point.

4. A minor point, but the definition of penetration depth varies, depending on whether power or field value is referred to. Here it’s field value and is consistent with ref (7) p 508 (86% absorbed or 14% remaining, with the field value 37% of original). This needs to be made clear. Incidentally, this may not hold for near-field conditions (see eq 6 on p 504 of ref (7)).

We better explained how we calculated the penetration depth, also according to the suggestion of the Reviewer #2

We calculated SAR depth of penetration according to the exponential law reported in [6] that is valid along the depth direction, from the skin (at the surface) to the deeper tissues in the head. For all our models this direction is along the z axis (see Figure 2). In details, we evaluated the SAR trend along the z axis from the axial (yz plane) SAR maps extracted from XFDTD results, considering the cell (pixel) with the maximum value of local SAR, for each model.

We better explained how we calculated SAR depth of penetration in a dedicated section “Data processing” in the Materials and Methods section (page 5 of 20).

 Grammar needs attention: for example:

P 1. The bands assigned to … includes frequencies below..’

P 2. ‘human hand to represent the smart-phone model.’\

‘simple tissue models estimation of the differences…’

P 3. Antenna patch was constituted of copper..

‘The antenna was inserted of about 2 mm..’ – meaning unclear

P 11. ‘the thorax was entirely interested…’, the genital area was also interested…’ – not clear what is meant by ‘interested’

P 15. ‘mobile terminal’ maybe ‘mobile device’

Fixed as suggested by the reviewer

Reviewer 2 Report

The paper reports a numerical dosimetry investigation of adults and children exposed to the electromagnetic field at 28 GHz generated by the antenna of a 5G mobile phone in three exposure scenarios: phone call, texting and browsing. The analysis is conducted on four anatomical human models from the Virtual Population: Duke (adult male), Ella (adult female), Billie (child female) and Thelonious (child male). The posture of the four human models is kept the same in all the exposure scenarios, which simply define the relative positioning of the mobile phone with respect to the body. The exposure scenarios can be seen as case studies in which the phone is moved further from the user: during the phone call the distance is just of 2 mm, during texting it is of about 16 cm, and during browsing it is of about 21 cm. The proposed analysis is interesting and useful, since it combines different models and different scenarios to provide an insight of the variability of the exposure within the population.

The results are mainly collected in terms of raw SAR distribution and maximum raw SAR, but this does not allow a direct understanding of the safety of the exposure scenario. A better understanding could be obtained providing the results in terms of temperature increase or with reference to the dosimetric quantities adopted by the relevant guidelines or standards. In order to increase the accessibility to the obtained results, the Authors should provide a relation between the computed raw SAR and the averaged quantities used by the relevant guidelines, motivating, also with reference to the local averages, the comment in the Discussion section “…we can conclude, with no doubt, that all the considered exposure scenarios comply with the safety limits relative to RF exposure…”.

The Authors should also address of the following issues:

  1. In section 2.3, it is written that a mesh with minimum size of 0.08 mm and maximum size of 1.071 mm is used. The minimum size is reasonable to account for the geometrical details of the antenna. The maximum size is motivated stating that it should be one-tenth of the radiation wavelength for an acceptable numerical accuracy. The wavelength at 28 GHz in vacuum is of about 10 mm, so the choice is reasonable in air, but in the biological tissues it is significantly less. For example, in the skin the wavelength at 28 GHz is about 2.6 mm, so the maximum resolution here should be of about 0.26 mm. In addition, the adopted resolution should be less than the penetration depth of the radiation, that in the skin at 28 GHz is equal to about 0.6 mm. Indeed, in the cited references 31 and 32 a resolution of 0.25 mm is used to discretize the human models for simulations at frequencies above 10 GHz. If the Authors use a coarser mesh, they should check the accuracy of the results with a mesh convergence study to corroborate their choice;
  2. It is not clear which sagittal, axial and coronal planes are pictured in figures 3, 7 and 8. I can guess that they are the planes in which the maximum SAR value is found, but it is not stated in the manuscript. Moreover, it is very difficult to understand which plane is actually pictured, especially in some cases like the sagittal planes in figure 3 or the coronal planes in figure 7. I suggest to denote in each image of the three figures the other two planes with dashed lines, or to use three-dimensional insets to illustrate the selected planes;
  3. Besides minor graphical choices, like the chromatic scale, I cannot see a substantial difference between figure 3 and figure 4. The Authors should remove figure 4 and make reference to figure 3 for the discussion of the results;
  4. A number of comments in the manuscript are devoted to the SAR depth of penetration, but it is quite vague how this quantity has been estimated. The direction along which the depth is evaluated should be defined unambiguously, like the direction perpendicular to the skin, but it does not seem to be the case. The values of SAR depth of penetration reported in table 3 and the trends plotted in figure 6 could be different one to each other simply because of the different relative angle between the red lines and the head models shown in figure 4. A precise definition of the SAR depth of penetration and a clear description of its evaluation should be added to the manuscript. Moreover, I suggest to use the decibel also in figure 6, so that only the trend in the four models can be compared, without biases due to the actual maximum values (which are already reported in table 2).

Some typos:

Page 4 of 15, section 2.3, “size” is repeated twice.

Page 5 of 15, section 2.3, “belonged” should be “belonging”.

Page 9 of 15, section 3.2, “of the model” is written in italic.

Page 11 of 15, section 3.3, “parameter had much lower values” is written in italic.

Page 11 of 15, section 3.3, “not more” should be “no longer”.

Author Response

The authors want to thank the reviewers for their precious comments about the manuscript.

The manuscript is substantially changed and it was rewritten. For this reason, the reviewers can find both a marked version and a clean version of the manuscript. Please, refer to the marked version for the point-by-point response to reviewers.

Reviewer 2

The results are mainly collected in terms of raw SAR distribution and maximum raw SAR, but this does not allow a direct understanding of the safety of the exposure scenario. A better understanding could be obtained providing the results in terms of temperature increase or with reference to the dosimetric quantities adopted by the relevant guidelines or standards. In order to increase the accessibility to the obtained results, the Authors should provide a relation between the computed raw SAR and the averaged quantities used by the relevant guidelines, motivating, also with reference to the local averages, the comment in the Discussion section “…we can conclude, with no doubt, that all the considered exposure scenarios comply with the safety limits relative to RF exposure…”.

As suggested by both the reviewers, we added in the manuscript the results of our simulations in terms of Absorbed Power Density (see new Table 2). We explained, in a dedicated section “Data processing” in the Materials and Methods section, how Sab has been calculated and reported the results in the new Table 2. We changed, in the Discussion section, our comment relative to the compliance of the simulated exposure conditions with the safety limits (page 17 of 20):

Since the very low values obtained for Sab (<2.15% of ICNIRP basic restriction in the phone call scenario, <0.16% of ICNIRP basic restriction in the writing scenario, and <0.03% of ICNIRP basic restriction in the browsing scenario) , we can conclude that all the considered exposure scenarios comply with the safety limits relative to RF exposure, both for adult and child models [4,7,8].

The Authors should also address of the following issues:

1. In section 2.3, it is written that a mesh with minimum size of 0.08 mm and maximum size of 1.071 mm is used. The minimum size is reasonable to account for the geometrical details of the antenna. The maximum size is motivated stating that it should be one-tenth of the radiation wavelength for an acceptable numerical accuracy. The wavelength at 28 GHz in vacuum is of about 10 mm, so the choice is reasonable in air, but in the biological tissues it is significantly less. For example, in the skin the wavelength at 28 GHz is about 2.6 mm, so the maximum resolution here should be of about 0.26 mm. In addition, the adopted resolution should be less than the penetration depth of the radiation, that in the skin at 28 GHz is equal to about 0.6 mm. Indeed, in the cited references 31 and 32 a resolution of 0.25 mm is used to discretize the human models for simulations at frequencies above 10 GHz. If the Authors use a coarser mesh, they should check the accuracy of the results with a mesh convergence study to corroborate their choice;

We thank the reviewer for this comment. We changed the paragraph to better explain the meshing process used (page 4 of 20):

An automatic non-uniform mesh was chosen with a minimum size of 0.08 mm and a maximum size of 1.071 mm. The maximum cell size was automatically chosen equal to one-tenth of the free space wavelength in compliance with the well-known rule to suppress the numerical dispersion error in FDTD simulations [34,35]. Using the intelligent meshing option, we set the minimum number of grid cells per wavelength equals to 15. The material properties of dielectric materials, such as human tissues, have been used to determine the wavelength. As result of this automatic meshing the human model’s discretization had a resolution of 0.15 mm in the region of interest. The adopted resolution permitted to obtain a good computational accuracy, also considering the penetration depth of radiation in the skin, that is about 1 mm at 28GHz.

2. It is not clear which sagittal, axial and coronal planes are pictured in figures 3, 7 and 8. I can guess that they are the planes in which the maximum SAR value is found, but it is not stated in the manuscript. Moreover, it is very difficult to understand which plane is actually pictured, especially in some cases like the sagittal planes in figure 3 or the coronal planes in figure 7. I suggest to denote in each image of the three figures the other two planes with dashed lines, or to use three-dimensional insets to illustrate the selected planes;

Figures 3, 7 and 8 has been changed: we added, for each model in all three scenarios, a three-dimensional inset to illustrate the selected planes and we denoted in each image the other two planes with dashed lines, as the reviewer suggested.

Please, note that Figure 7 now is Figure 6 and Figure 8 now is Figure 7, since Figure 4 has been removed as suggested by the reviewer in the following point.

3. Besides minor graphical choices, like the chromatic scale, I cannot see a substantial difference between figure 3 and figure 4. The Authors should remove figure 4 and make reference to figure 3 for the discussion of the results;

We removed figure 4 as suggested by the reviewer.

4. A number of comments in the manuscript are devoted to the SAR depth of penetration, but it is quite vague how this quantity has been estimated. The direction along which the depth is evaluated should be defined unambiguously, like the direction perpendicular to the skin, but it does not seem to be the case. The values of SAR depth of penetration reported in table 3 and the trends plotted in figure 6 could be different one to each other simply because of the different relative angle between the red lines and the head models shown in figure 4. A precise definition of the SAR depth of penetration and a clear description of its evaluation should be added to the manuscript.

We calculated SAR depth of penetration according to the exponential law reported in [6] that is valid along the depth direction, from the skin (at the surface) to the deeper tissues in the head. For all our models this direction is along the z axis (see Figure 2). In details, we evaluated the SAR trend along the z axis from the axial (yz plane) SAR maps extracted from XFDTD results, considering the cell (pixel) with the maximum value of local SAR, for each model.

We better explained how we calculated SAR depth of penetration in a dedicated section “Data processing” in the Materials and Methods section (page 5 of 20).

5. Moreover, I suggest to use the decibel also in figure 6, so that only the trend in the four models can be compared, without biases due to the actual maximum values (which are already reported in table 2).

As suggested by the reviewer, to permit a better comparison of the trend in the four models without biases due to the maximum values, we changed the Figure 6 (now Figure 5). However, we used normalized value (with a scale between 0 and 1) instead of decibel, to show the exponential decay from the surface to deeper regions of the local SAR. This permit a better comparison of our results with the ones shown in literatures (see, for example, References 6)

Some typos:

Page 4 of 15, section 2.3, “size” is repeated twice.

Page 5 of 15, section 2.3, “belonged” should be “belonging”.

Page 9 of 15, section 3.2, “of the model” is written in italic.

Page 11 of 15, section 3.3, “parameter had much lower values” is written in italic.

Page 11 of 15, section 3.3, “not more” should be “no longer”.

Fixed as suggested by the reviewer

Round 2

Reviewer 2 Report

All the raised issues have been solved, so I think that the manuscript is now suitable for publication.